# Ketamine Increases Proliferation of Human iPSC-Derived Neuronal Progenitor Cells via Insulin-Like Growth Factor 2 and Independent of the NMDA Receptor

**DOI:** 10.3390/cells8101139

**Published:** 2019-09-24

**Authors:** Alessandra Grossert, Narges Zare Mehrjardi, Sarah J. Bailey, Mark A. Lindsay, Jürgen Hescheler, Tomo Šarić, Nicole Teusch

**Affiliations:** 1Bio-Pharmaceutical Chemistry and Molecular Pharmacology, Faculty of Applied Natural Sciences, Technische Hochschule Köln, 51373 Leverkusen, Germany; alessa.grossert@web.de; 2Center for Physiology and Pathophysiology, Institute for Neurophysiology, Medical Faculty, University of Cologne, 50931 Cologne, Germany; nargeszare.mehrjardi@med.uni-duesseldorf.de (N.Z.M.); j.hescheler@uni-koeln.de (J.H.); tomo.saric@uni-koeln.de (T.Š.); 3Department of Pharmacy and Pharmacology, University of Bath, Bath BA2 7AY, UK; sb304@bath.ac.uk (S.J.B.); mal37@bath.ac.uk (M.A.L.); 4Department of Biomedical Sciences, Faculty of Human Sciences, University of Osnabrück, 49076 Osnabrück, Germany

**Keywords:** human iPSC-derived NPCs, depression, neurogenesis, ketamine, IGF2, cAMP, p11

## Abstract

The N-methyl-D-aspartate (NMDA) receptor antagonist ketamine offers promising perspectives for the treatment of major depressive disorder. Although ketamine demonstrates rapid and long-lasting effects, even in treatment-resistant patients, to date, the underlying mode of action remains elusive. Thus, the aim of our study was to investigate the molecular mechanism of ketamine at clinically relevant concentrations by establishing an in vitro model based on human induced pluripotent stem cells (iPSCs)-derived neural progenitor cells (NPCs). Notably, ketamine increased the proliferation of NPCs independent of the NMDA receptor, while transcriptome analysis revealed significant upregulation of insulin-like growth factor 2 (IGF2) and p11, a member of the S100 EF-hand protein family, which are both implicated in the pathophysiology of depression, 24 h after ketamine treatment. Ketamine (1 µM) was able to increase cyclic adenosine monophosphate (cAMP) signaling in NPCs within 15 min and cell proliferation, while ketamine-induced IGF2 expression was reduced after PKA inhibition with cAMPS-Rp. Furthermore, 24 h post-administration of ketamine (15 mg/kg) in vivo confirmed phosphorylation of extracellular signal-regulated protein kinases 1 and 2 (ERK1/2) in the subgranular zone (SGZ) of the hippocampus in C57BL/6 mice. In conclusion, ketamine promotes the proliferation of NPCs presumably by involving cAMP-IGF2 signaling.

## 1. Introduction

According to the World Health Organization (WHO), depression is the leading cause of disability worldwide, with more than 300 million patients affected [1]. Currently available antidepressants, such as selective serotonin reuptake inhibitors (SSRIs) or tricyclic antidepressants (TCAs) have a delayed onset of action and show antidepressant effects only in one-third of patients, indicating the urgent need for the development of more effective medications [2,3].

Chronic stress is a risk factor for major depressive disorder (MDD), which is associated with impaired neuroplasticity and suppression of hippocampal neurogenesis [4]. Several studies have reported decreased hippocampal volume in depressed individuals correlating with reduced cell proliferation and impaired function of the adult hippocampus [4,5,6].

Conversely, chronic antidepressant treatment has been shown to increase the proliferation of hippocampal progenitor cells in rodents and humans [5,7]. For instance, antidepressant treatment with fluoxetine (SSRI) significantly enhanced the number of proliferating cells in the dentate gyrus of the hippocampus in rats after 14 and 28 days of treatment [8]. Moreover, a post-mortem study of human brain tissue showed that MDD patients, who received antidepressant treatment with SSRIs or TCAs for three months prior to death had a higher number of NPCs and 50% more dividing cells in the subgranular zone (SGZ) of the dentate gyrus than unmedicated subjects [9]. Therefore, enhanced neural progenitor cell (NPC) proliferation is suspected to stimulate neurogenesis, thereby supporting the antidepressant response [5,10].

Emergent research has focused on gaining molecular insights in processes responsible for neural proliferation, potentially providing novel drug targets for the therapy of MDD [7]. In this context, recent evidence displays insulin-like growth factor 2 (IGF2)as a critical target for psychiatric diseases by playing a pivotal role in neurogenesis and in the antidepressant responses of the most widely used classical antidepressant drugs. IGF2 is abundantly expressed in the hippocampus, and decreased expression is recovered by TCA treatment in mice 21 days after daily treatment [11,12,13,14]. In addition, the multifunctional protein p11 (also known as S100A10) is of growing interest in the therapy of mood disorders. Similar to IGF2, expression of p11 is reduced in the hippocampus of patients suffering from depression as well as in disease-relevant animal models. Accordingly, chronic antidepressant therapy has been shown to enhance expression of p11 in mice after 14 days of treatment, and moreover, p11 is involved in neurogenic effects presumably through induction of the brain-derived neurotrophic factor (BDNF)-tropomyosin receptor kinase B (TrkB)cascade, a key signaling pathway in the therapeutic response of antidepressants [15,16,17]. Furthermore, growing evidence suggests the involvement of the cyclic adenosine monophosphate (cAMP) cascade in the pathology of depression. A study comparing cAMP activity in unmedicated versus medicated patients suffering from depression demonstrated impaired cAMP signaling, as indicated by 18% lower 11C-(R)-rolipram binding across 10 preselected regions of the brain. Indeed, antidepressant treatment with SSRIs increased the cAMP activity significantly by 12 ± 36% but failed to induce symptom improvement after two months of treatment, correlating with the delayed onset of action of classical antidepressant drugs [18].

Interestingly, the N-methyl-D-aspartate (NMDA) receptor antagonist ketamine, which has been approved for anesthesia in 1970, shows after a single infusion, rapid (30 min) and long-lasting (seven days) antidepressant effects in subanesthetic doses, even in treatment-resistant patients. It is currently speculated, that the antidepressant effect is not exclusively mediated by its NMDA-antagonism, as other NMDA-receptor antagonists such as memantine failed to induce rapid antidepressant effects in clinical trials [19]. Despite the elusive molecular mode of action, the S-enantiomer of ketamine (esketamine) has been recently approved for treatment-resistant depression by the US Food and Drug Administration as the first antidepressant of a novel class [20].

To date, various signaling pathways involved in ketamine’s putative role as an antidepressant have been described. For instance, ketamine-dependent stimulation of α-amino-3-hydroxy-5-methylisoxazole-4-propionate (AMPA) receptors by phosphorylation of the glutamate ionotropic receptor AMPA type subunit 1 (GluA1) subunit and activation of the mammalian target of rapamycin (mTOR) lead to the induction of the BDNF-TrkB pathway, thereby inducing rapid and sustained antidepressant effects [10,21,22]. As ketamine is known to be involved in a wide range of more signaling pathways, such as, for instance, opioid potentiation, release of neuromodulators like dopamine or noradrenaline, or the regulation of metabotropic glutamate receptors (mGluR), the precise molecular signal transduction still needs to be explored [20]. Furthermore, recent publications indicate the induction of hippocampal neurogenesis in rodents after ketamine treatment, suggesting the involvement of cell proliferation in the described striking antidepressant effects [10,23]. More precisely, acute ketamine treatment (7 mg/kg) showed hippocampal activation of pERK1/2 in mice after 6 and 24 h and, moreover, significantly enhanced 5-bromo-2′-deoxyuridine positive (BrdU+) cells seven days after administration [10].

However, despite significantly increasing research activities on ketamine during the last years, the precise molecular mechanism responsible for its unique antidepressant effects remains controversial. As the pathophysiology of depression correlates with decreased adult neurogenesis, we aimed to investigate the molecular effects of ketamine on neural progenitor cell proliferation using a human-based induced pluripotent stem cell (iPSC)-model. The serum levels of ketamine in patients receiving the most common antidepressant dose of 0.5 mg/kg for 40 min intravenous (i.v.) led to a maximal plasma concentration of 185 ng/mL corresponds to ≅ 0.78 µM [24].

In the present study, ketamine-induced proliferation of human iPSC-derived NPCs and bioinformatic analysis of RNA-seq data revealed increased p11 and IGF2 expression 24 h after ketamine treatment. In line with this, ketamine dependent proliferation was significantly impaired after IGF2 knockdown. Moreover, ketamine was able to enhance cAMP signaling in NPCs, and both, cell proliferation as well as IGF2 expression, were reduced after protein kinase A (PKA)-inhibition. Noteworthy, our Nestin-expressing NPCs do not express functional NMDA receptors, suggesting that the pro-proliferative effect of ketamine in NPCs is NMDA receptor-independent.

## 2. Materials and Methods

### 2.1. Chemicals and Reagents

5-Bromo-2′-deoxyuridine (BrdU), (±)-Epinephrine hydrochloride, A23187, glutamate, forskolin, isobutyl-1-methylxanthine (IBMX), 4-(3-Butoxy-4-methoxybenzyl) imidazolidin-2-one (Ro 20-1724) (±)-ketamine hydrochloride and N-Methyl-D-aspartic acid (NMDA) were purchased from Sigma–Aldrich. Triethylammonium salt (cAMPS-Rp) was obtained from Tocris Bioscience, and Quest Fluo-8^TM^ AM from AAT Bioquest. All compounds were dissolved in dimethyl sulfoxide (DMSO; Carl Roth). Recombinant human IGF2 protein was purchased from R&D Systems and reconstituted in phosphate-buffered saline (PBS).

### 2.2. Cell Culture

Human iPSC-derived IMR90 NPCs and Ro-iPSC (Royan iPSC) NPCs were generated by Dr. Tomo Šarić (The University of Cologne, Institute for Neurophysiology, Cologne, Germany) [25]. Cells were maintained as monolayer cultures on poly-L-ornithine/laminin (Sigma–Aldrich) coated six-well plates in serum-free neural stem cell medium containing dulbecco’s modified eagle medium/ nutrient mixture F-12 (DMEM/F12) glutamax (Gibco), 1× N2 supplement (Gibco), 1.6 mg/L D-glucose (Sigma–Aldrich), 20 µg/mL insulin (Sigma–Aldrich), 1 µL/mL B27 (Gibco), 20 ng/mL fibroblast growth factor-basic (bFGF) (Peptrotech), 20 ng/mL human epidermal growth factor (hEGF) (Sigma–Aldrich) [26]. The medium was changed every other day. The cells were maintained in a humidified atmosphere at 37 ºC, and 5% CO_2_ and subcultured when confluency reached 90%.

### 2.3. Real-Time Cell Proliferation

NPCs were seeded at a density of 8 × 10^4^ cells per ml in poly-L-ornithine/laminin (Sigma–Aldrich) coated 96-well image-lock plates (Sartorius, Goettingen, Germany) 24 h prior to the experiment. Images were taken every hour for a time frame of three days with the IncuCyte^®^ Zoom from Sartorius (Goettingen, Germany) using a 10× objective. Confluency of cells was determined as Cell-Body Cluster Area [Phase] (mm^2^/mm^2^) using the IncuCyte^®^ NeuroTrack Software (version 2016B). For protein kinase A (PKA) inhibition studies, cells were pretreated with 1 µM triethylammonium salt (cAMPS-Rp; Tocris Bioscience) for 15 min at 37 °C, and 5% CO_2_ before ketamine or DMSO control was added.

### 2.4. Immunofluorescence Staining

NPCs were seeded at a density of 3 × 10^4^ cells per ml in poly-L-ornithine/laminin (Sigma–Aldrich) coated black 96-well clear-bottom plates (Greiner Bio-One) and allowed to attach overnight. Cells were fixed and permeabilized with pre-chilled methanol at −20 °C for 10 min. After blocking with 0.5% fish gelatin (Sigma–Aldrich) in PBS, cells were incubated with the primary antibody overnight at 4 °C, followed by secondary antibody (1:100) for 2 h at room temperature. Primary antibodies used were as follows: mouse-anti-Nestin (1:100, Santa Cruz biotechnology, Dallas, TX, USA, sc-23927), mouse-anti-TrkB (1:100, Santa Cruz Biotechnology, sc-377218), mouse-anti-Pax6 (1:400, Abcam, Cambridge, MA, USA, ab5790), and rabbit-anti-GluA1 (1:100, Merck Millipore, Darmstadt, Germany, A1504). Secondary antibodies were purchased from Santa Cruz Biotechnology (mouse anti-rabbit IgG-FITC (fluorescein isothiocyanate), sc-2359; goat anti-rabbit IgG-Rhodamine, sc-2091; mouse IgG kappa binding protein (m-IgGκ BP-CFL) 488, sc-516176). For nuclear staining, cells were incubated with 4′,6-Diamidin-2-phenylindol (DAPI, Sigma–Aldrich) for 15 min at room temperature at a concentration of 3 µM. Images were taken with the Axiovert 200M fluorescence microscope (Zeiss, Jena, Germany).

### 2.5. Fluorescence-Based Calcium Mobilization Assay

Kinetic measurements of transient intracellular calcium mobilization in NPCs were performed in poly-L-ornithine/laminin (Sigma–Aldrich) coated black 96-well clear-bottom plates (Greiner Bio-One). Cells were seeded at a density of 3.0 × 10^5^ cells per ml and incubated for 24 h. The following day, the medium was removed and cells were loaded with 4 µM Fluo-8 (AAT Bioquest) as the indicator dye, reconstituted in a modified Tyrode’s assay buffer (10 mM 4-(2-hydroxyethyl)-1-piperazineethanesulfonic acid (HEPES), 135 mM NaCl, 5 mM KCl, 2.5 mM CaCl_2_, and 10 mM glucose adjusted to pH = 7.4.) and incubated for 30 min at 37 °C and 5% CO_2_, and another 30 min at room temperature in the dark [27,28]. Then, the Fluo-8 loading buffer was removed and replaced with fresh Tyrode’s assay buffer. The agonists, diluted in Tyrode’s assay buffer, were added automatically using the injector unit from Tecan Infinite M1000 Pro microplate reader Tecan (Maennedorf, Switzerland). After 16 s of baseline measurements, the compound was added, and fluorescence was measured for 80 s using excitation at λ = 490 nm and emission at λ = 525 nm. Responses were measured as the maximal peak height in relative fluorescent units (RFUs), and the maximum fluorescence signal was generated with the calcium ionophore A23187 (Sigma–Aldrich).

### 2.6. BrdU Analysis of Cell Proliferation In Vitro

NPCs were seeded at a density of 8 × 10^4^ cells per ml in poly-L-ornithine/laminin (Sigma–Aldrich, St. Louis, MO, USA) coated 96-well image-lock plates (Sartorius, Goettingen, Germany) 24 h prior to treatment. Cells were treated with vehicle (0.01% DMSO) or 1 µM ketamine and incubated for 24 h at 37 °C, 5% CO_2_. The culture medium was removed then and replaced with 10 µM BrdU labeling solution dissolved in the culture medium. After an incubation time of 4 h (37 °C, 5% CO_2_), the labeling solution was removed and cells were rinsed five times with Phosphate Saline Buffer (PBS, pH 7.4).

For BrdU immunofluorescence staining, cells were fixed with 4% paraformaldehyde (Sigma-Aldrich) dissolved in PBS for 30 min at room temperature. After permeabilization with 0.1% Triton™ X-100 in PBS for additional 5 min, cells were blocked with 5% bovine serum albumin (BSA) in PBS for 30 min at room temperature and washed with 0.1% Triton™ X-100 for 5 min. For DNA hydrolysis cells were incubated with 2 M HCl for 1.5 h at room temperature. To neutralize the acid, cells were incubated with 0.1 M sodium borate buffer (pH 8.5). After washing with 0.1% Triton™ X-100, cells were incubated with the primary mouse monoclonal BrdU antibody (1:500, Santa Cruz Biotechnology, sc-32323) overnight at 4 °C in blocking solution. The following day, samples were incubated with secondary antibody (1:100, Santa Cruz biotechnology, sc-516176) for 2 h at room temperature. For nuclear staining, cells were incubated with 4′,6-Diamidin-2-phenylindol (DAPI, Sigma Aldrich) for 15 min at room temperature at a concentration of 3 µM. Images were taken with the Axio Vert.A1 microscope (Zeiss, Oberkochen, Germany). BrdU incorporation was quantified by the calculation of BrdU+ cells as a percentage of the total number of DAPI-labeled nuclei using ImageJ (Fiji, v1.52o) for image analysis [29].

### 2.7. RNA Extraction and Library Preparation for Transcriptome Sequencing

IMR90 NPCs were seeded at a density of 1 × 10^6^ cells per ml in poly-L-ornithine/laminin (Sigma–Aldrich) coated six-well plates and allowed to attach overnight. The following day, cells were treated with 1 µM ketamine or vehicle for 24 h. For all samples, total RNA was isolated using the RNeasy Mini kit from Qiagen, including an on-column DNAase treatment (15 min at room temperature, Qiagen) according to the manufacturer’s instructions. The RNA concentration was determined using the NanoQuant Plate™ (Tecan, Salzburg, Austria). All library preparations were sequenced on an Illumina Hiseq platform with a paired-end read length of 125 bp/150 bp. The library construction and sequencing were performed by Novogene (HK) Company Limited (Wan Chai, Hong Kong).

### 2.8. RNA-Seq Data Analysis

The paired-end reads were aligned to the human reference genome (hg38) using Hisat2 (version 2.0.4) using the following command-line options: hisat2 −q–dta–rna-strandness FR −x < reference-genone.gtf > −1 < forward_strand.fa > −2 < reverse-strand file.fa > −S < output.sam > . Output SAM files were then sorted and converted to BAM files (samtools sort −@ 8 −o output.bam output.sam) and indexed (samtools index −b output.bam) in Samtools [30,31,32]. The profile of gene expression (using the Gencode v27 database) in the BAM files for each sample were determined using Stringtie: stringtie < sample.BAM > −G < GenCodev26.gtf > −1o < samples.gtf > −e −A < sample.txt > and is reported as FPKM (Fragments Per Kilobase Million) [33]. Following feature counting: featureCounts -a < reference-genome.gtf > −g gene_name -o counts.txt Control_*.bam IPF_*.bam the differential gene expression was assessed using DeSeq2 and the following R script: curl −s -O http://data.biostarhandbook.com/rnaseq/code/deseq 2.r cat simple_counts.txt | Rscript deseq2.r 3x3 > results_deseq2.txt [34]. Genes with an adjusted p-value < 0.05 were assigned as differentially expressed.

### 2.9. Quantitative Real-Time PCR

To determine IGF2 and p11 expression after ketamine treatment and to confirm RNA-Seq data, aliquots of the non-pooled RNA samples were used. The following primers were purchased from Eurofins, Ebersberg, Germany: GAPDH forward: (TGCACCACCAACTGCTTAGC), GAPDH reverse: (GGCATGGACTGTGGTCATGAG), IGF2 forward: (CCAAGTCCGAGAGGGACGT), IGF2 reverse: (TTGGAAGAACTTGGCCACG), p11 forward: (ACCACACCAAAATGCCATCT), p11 reverse: (CTGCTCATTTCTGCCTACTT). Total RNA was extracted with the RNeasy Mini kit from Qiagen (Hilden, Germany) according to the manufacturer’s guideline and reverse transcriptase PCR was performed using the \Reverse Transcription Kit (Promega). Real-time PCR was conducted with Quantitect SYBR Green from Qiagen based on the following protocol: Preincubation at 95° for 900 s, amplification was performed over 45 cycles (95 °C for 15 s, 55 °C for 25 s, and 72 °C for 10 s). Expression values were calculated according to the 2^−ΔΔCT^ method [35]. The sample values were normalized to the housekeeping gene GAPDH (glyceraldehyde 3-phosphate dehydrogenase).

### 2.10. siRNA Transfection

Lipofectamine Stem Reagent (Invitrogen) and Opti-MEM I reduced serum medium (Invitrogen) were used for the transfection experiments according to the manufacturer’s specifications. 1 × 10^6^ cells were transfected with 2 µM IGF2 (h) siRNA (Santa Cruz Biotechnology) or Silencer^®^ Select Negative Control No. 2 siRNA (Invitrogen). After 24 h, the medium was replaced with fresh medium and incubated for another 24 h, before cells were harvested and expression upon knockdown of interest was analyzed using quantitative real-time PCR.

### 2.11. Cyclic Adenosine Monophosphate (cAMP)-Glo Assay

The cAMP-Glo assay was obtained from Promega (Madison, WI, USA) and performed according to the manufacturer’s protocol. Briefly, 1 × 10^5^ cells per ml were seeded in poly-L-ornithine/laminin (Sigma–Aldrich) white 96-well plates (Greiner Bio-One, Kremsmuenster, Austria) and allowed to adhere overnight (37 °C, 5% CO_2_). Cells were treated in induction buffer (500 µM 3-Isobutyl-1-methylxanthin (IBMX) and 100 µM Ro 20-1724 in PBS) with 1 µM ketamine for 15 min, and forskolin was used as a positive control as it activates the adenylyl cyclase enzyme and increases intracellular cAMP levels. Luminescence was determined using Tecan Infinite 200 pro reader (Tecan Group AG, Männedorf, Switzerland).

### 2.12. In Vivo Administration of Ketamine and Immunohistochemistry of BrdU and pERK1/2

In vivo experiments were conducted at the University of Bath, UK. All experiments were performed in accordance with UK Home Office guidelines, including local ethical review, and the Animals (Scientific Procedures) Act1986 incorporating European Union Directive 2010/63/EU. Adult male C67BL/6 mice (8–9 weeks old) were used to study the proliferative effects of ketamine in vivo. Mice were originally from Charles River (C57BL/6NCrl) and bred in house at the University of Bath for more than 10 years. Mice were randomly assigned to treatment groups (n = 4 per group) and injected intraperitoneally (i.p.) with 15 mg/kg ketamine, a dose established to be sufficient to induce antidepressant effects in C57BL/6 mice (Lalji and Bailey, unpublished), or 0.9% saline as a control. To assess the lasting effects of ketamine on proliferation, 24 h after drug treatment, mice were injected i.p. with BrdU (200 mg/kg) (Sigma-Aldrich). Two hours later, mice were anesthetized with 100 mg/kg sodium pentobarbital and perfused transcardially with 4% paraformaldehyde (PFA) in PBS. Brains were post-fixed overnight in 4% PFA at 4 °C. The following day, 40 µm free-floating coronal sections of the brains were taken using a vibratome (Intracell 1000plus Sectioning System) (every 4^th^ section of the hippocampus per mouse) and transferred to 24-well plates loaded with 500 µL PBS and stored overnight at 4 °C. To denature DNA for BrdU staining, sections were incubated with 2 M HCl for 15 min at 37 °C. After neutralization with PBS, sections were blocked with 5% donkey serum in PBS for 1 h at room temperature and incubated with BrdU primary antibody diluted in blocking solution (1:500, Abcam, ab6326) overnight at 4 °C. The following day, sections were rinsed with PBS and incubated with secondary antibody diluted in PBS containing 0.3% Triton X (1:1000, Abcam, ab150153) for 2 h at room temperature. Sections were mounted on slides with DAPI-containing mounting medium (Vectashield, Biozol, Eching, Germany). Images were taken using a Leica DMI4000B inverted wide-field fluorescent microscope at 50× and 200× magnification. Quantification of BrdU+ cells was performed on a random series of twelve sections throughout the entire hippocampus with the experimenter blinded to treatment using Image J (Fiji) [29].

For further microscopic analysis, sections were stored in PBS with 0.01% sodium azide and shipped to University of Osnabrück, Germany. To evaluate phosphorylation of ERK1/2, sections were incubated overnight at 4 °C with pERK1/2 primary antibody (1:400 in PBS, Cell Signaling Technology, #4370) and secondary anti-rabbit antibody (1:100, Santa Cruz Biotechnology, sc-2359) for 2 h. Images were taken using the Zeiss LSM 880 Airyscan system at 63× magnification.

### 2.13. Statistical Analysis

All data shown represent at least three independent experiments. Error bars show ± SEM of all the means of triplicate values. All statistical analysis and graphs were generated with GraphPad Prism v. 7.02 (Graphpad Software, San Diego, CA, USA) using one-way ANOVA and the underlying Sidak’s multiple comparisons test or unpaired two-tailed t-test for the comparison of two groups.

*p* < 0.05 was chosen to define statistically significant differences. In all figures, one-star represents a significance of *p* < 0.05, two stars of *p* < 0.01, three stars of *p* < 0.001, four stars *p* < 0.0001, and “ns” means not significant.

## 3. Results

### 3.1. Characterization of Undifferentiated iPSC-Derived NPCs Proves Absence of Ionotropic Glutamate Receptors

RNA-Seq analysis of the read counts (fragments per kilobase of transcript per million mapped reads, FPKM) indicates that NPCs highly express the neuronal markers Sox2 (<2400 FPKM) and Nestin (<18,500 FPKM), while of the respective read counts for ionotropic glutamatergic receptors GluA1 (<750 FPKM) and GluN2B (<190 FPKM) were significantly lower or even below the detection level (GluN1 (<1 FPKM), Figure 1A). To confirm the expression of the characteristic neuronal progenitor markers on protein levels, NPCs were stained for Nestin and Pax6 (Figure 1B). As expected, nearly all cells were positive for NPCs markers, demonstrating a homogeneous population of neural progenitor cells. Analysis of protein expression of distinct receptors reported to be involved in the molecular effects of ketamine revealed that NPCs showed expression of the BDNF receptor TrkB, but no signal for the glutamate ionotropic receptor subunit GluA1 (Figure 1B). Notably, six weeks of differentiation led to neurons positive for the α-amino-3-hydroxy-5-methylisoxazole-4-propionate (AMPA) receptor subunit GluA1 (Appendix A). Noteworthy, a second cell line (Ro-iPSC NPCs) showed similar neuronal progenitor characteristics like IMR90 NPCs (Appendix A).

To further confirm the absence of glutamate ionotropic receptors in undifferentiated NPCs, transient mobilization of intracellular calcium was analyzed by stimulation with agonists for the NMDA-receptor (NMDA and glutamate). As expected, upon stimulation with NMDA (10 µM) or glutamate (10 µM), no calcium mobilization was observed in either IMR90 or Ro-iPSC NPCs, while the calcium ionophore A23187 served as a positive control (Figure 1C; Appendix A).

### 3.2. Ketamine Increases Cell Proliferation of Human iPSC-Derived NPCs

We examined the effect of the NMDA receptor antagonist on cell proliferation in human iPSC-derived NPCs using the IncuCyte^®^ Zoom live-cell imaging system. Cells were imaged every hour over a time range of 72 h, and confluency of cells were calculated as the Cell-Body Cluster Area (Figure 2A). Noteworthy, after 72 h, ketamine was able to increase cell proliferation significantly by 38% compared to DMSO control (One-way ANOVA, posthoc *t*-test *p* < 0.05) (Figure 2B).

Next, we aimed to identify the most effective and clinically relevant concentration of ketamine on proliferation. As mentioned earlier, serum levels of ketamine in patients receiving the most common antidepressant dose of 0.5 mg/kg for 40 min i.v., increased up to a maximal plasma concentration of 185 ng/mL, which corresponds to ≅ 0.78 µM [24]. As the TrkB agonist BDNF has been described previously to increase NPC proliferation [36], BDNF served as a positive control. To further investigate whether the effects of ketamine on proliferation are concentration-dependent, we evaluated a concentration range starting at 0.5 µM up to 10 µM. While 0.5 and 1 µM ketamine both enhanced cell proliferation by 34% and 25%, respectively, 10 µM ketamine showed no effect compared to 0.01% DMSO control after 72 h (Figure 3A,B). Thus, we decided on the clinically relevant concentration of 1 µM for the following in vitro experiments. Importantly, ketamine did not induce proliferation of cancer cell lines (human cervix epithelioid carcinoma cells (“HeLa cells”), indicating a specific effect on neuronal progenitor cells (Appendix A).

To further confirm the proliferative influence of ketamine and to demonstrate that the ketamine dependent induction of NPC proliferation is a representative principle for progenitor cells derived from different individuals, we analyzed BrdU incorporation in a second NPC cell line (Ro-iPSC) in parallel. NPCs were treated for 24 h with ketamine or 0.01% DMSO control and were incubated for an additional 4 h with 10 µM BrdU to label proliferating cells. Notably, ketamine significantly increased the number of BrdU+ cells within 24 h in both cell lines (IMR90 NPCs by 18% and Ro-iPSC NPCs by 32% compared to the respective DMSO control. (Figure 4A,B).

### 3.3. Ketamine Alters Gene Expression in Human iPSC-Derived NPCs

As both NPC cell lines do not express functional NMDA receptors, our data reveal the pro-proliferative effect of ketamine in NPCs to be presumably NMDA receptor-independent (Figure 1A,B). To further investigate the molecular mechanism of action responsible for ketamine-dependent proliferation in human NPCs, we examined the effects at the transcriptional level. To this end, IMR90 NPCs were treated with vehicle (0.01% DMSO) or 1 µM ketamine for 24 h and total RNA was analyzed using RNA-Sequencing. NPCs displayed multiple stem/progenitor cell genes such as Nestin, Sox2, and Pax6 and ketamine treatment resulted in the significant up-regulation and down-regulation of 31 genes and 29 genes, respectively (Figure 5A,B). The upregulated genes IGF2 and p11 (S100A10), which are both implicated in the pathophysiology of depression [13,15], were selected to confirm the obtained RNA-Seq results by qPCR using aliquots of the non-pooled RNA samples (*n* = 3). qPCR validation showed a significant upregulation of IGF2 by 1.5-fold and p11 by 4.0-fold (Figure 5C) after 24 h of ketamine (1 µM) treatment. Moreover, the time course of IGF2 mRNA induction revealed a maximum upregulation after 24 h of ketamine treatment (Appendix A).

### 3.4. IGF2 Signaling is Involved in the Proliferative Effects of Ketamine

As IGF2 has been described as a regulator of hippocampal adult neurogenesis [12], the effect of a knockdown of IGF2 on NPC was studied. A reduction of IGF2 expression by 66% was achieved as confirmed by qPCR (Figure 6A). Knocking down IGF2 reduced cell proliferation of NPCs compared to noncoding (nc) siRNA controls by 18% and reduced ketamine’s proliferative effects by 17% (Figure 6B,C) compared to the quantification of BrdU+ cells after 24 h of treatment with 1 µM ketamine or 0.01% DMSO control. Furthermore, the addition of 50 ng/mL IGF2 induced cell proliferation and significantly prevented the reduction of cell proliferation upon IGF2 knockdown (Figure 6C). In conclusion, IGF2 regulates NPC cell proliferation and is involved in ketamine-induced proliferation.

### 3.5. Ketamine Increases cAMP Levels in NPCs within 15 min

Recent studies have shown a down-regulation of cAMP signaling in patients with major depressive disorder (MDD) and subsequent recovery by antidepressant treatment [18]. It has been reported previously that ketamine treatment resulted in an NMDA receptor-independent induction of cAMP signaling in C6 glioma cells, which, in turn, increased BDNF expression [37]. As ketamine’s effects in the iPSC-derived NPCs seem to be independent of NMDA receptors, we aimed to determine whether ketamine is able to increase cAMP in our NPC model. Based on the experiments of Wray et al. (2018), we treated IMR90 NPCs with 1 µM ketamine for 15 min and determined cAMP accumulation using the cAMP-Glo assay (Promega), where luminescence is inversely proportional to cAMP levels. Forskolin (activator of adenylyl cyclase enzyme) and epinephrine (an agonist of adrenergic receptors) were used as positive controls. In NPCs, ketamine-induced cAMP accumulation within 15 min by 15% (Figure 7A).

Next, cells were treated with the protein kinase A (PKA)-inhibitor cAMPS-Rp (1 µM) to inhibit cAMP signaling and the effects on IGF2 expression (Figure 7B), as well as on cell proliferation (Figure 7C) were examined. Whereas 1 µM cAMPS-Rp treatment alone had no effect on IGF2 expression and proliferation, ketamine’s effects on both, IGF2 expression and cell proliferation, were significantly (by 26% and 14%, respectively) reduced after PKA-inhibition.

### 3.6. Phosphorylation of ERK1/2 is Enhanced by Ketamine in the SGZ of Mice

To investigate whether ketamine-induced proliferation in human iPSC-derived NPCs correlates with effects in vivo, we examined the lasting effect of a single ketamine injection on hippocampal cell proliferation and extracellular signal-regulated protein kinases 1 and 2 (ERK1/2) phosphorylation in adult C57BL/6 mice 24 h post-administration (Figure 8A) as ERK1/2 is a well-described key regulator of proliferation in Nestin-positive neuronal progenitors separated from rat embryonic hippocampus [38]. Mice were treated with 15 mg/kg ketamine, a dose established to be sufficient to induce antidepressant effects in C57BL/6 mice (Lalji and Bailey, unpublished). Importantly, immunocytochemical analysis of phospho-ERK1/2 in the hippocampal subgranular zone (SGZ) revealed an increase in ketamine-treated mice compared to saline controls (Figure 8B). In line with Ma et al. [10], the quantification of BrdU+ cells did not display a significant effect on cell proliferation within the selected experimental time frame of 24 h ketamine treatment (Figure 8C,D).

In summary, our findings lead us to hypothesize that ketamine increases proliferation of NPCs via cAMP-IGF2 signaling (Figure 9).

## 4. Discussion

Proliferation of NPCs in the human hippocampus plays an essential role in antidepressant therapy, but the delayed onset of action is the major limitation of classical antidepressant drugs such as SSRIs or TCAs [8,39,40]. Here, we identified proliferative effects and first hints towards the underlying molecular mechanism of the rapid-acting and long-lasting antidepressant ketamine in human NPCs. The undifferentiated human iPSC-derived NPCs used in our study displayed typical characteristics of hippocampal NPCs by upregulating neuronal progenitor markers such Nestin and Pax6 while missing expression of functional ionotropic glutamate receptors, such as NMDA or AMPA receptors, which is in accordance with the previously described characterization of hippocampal human undifferentiated NPCs [9,28,39].

Regarding the molecular mode of action responsible for ketamine’s antidepressant effects, contradictory results have been reported. Notably, the role of the NMDA receptor is still a matter of debate as other NMDA receptor antagonists such as memantine failed to induce rapid and long-lasting antidepressant effects [19,37,41].

In our study, ketamine was able to increase the proliferation of human NPCs over a time-range of 72 h independent of the NMDA receptor. Several studies have already shown that chronic antidepressant treatment leads to increasing hippocampal neurogenesis and enhanced proliferation of NPCs in rodents and humans [4,5,8,39]. Moreover, ketamine has been recently reported to promote hippocampal neurogenesis in mice by inducing pERK1/2 activation (six and 24 h after administration) and enhancement of NPC proliferation seven days after treatment [10]. Consistent with these findings, we found enhanced phosphorylation of ERK1/2 in the SGZ of the hippocampus in mice 24 h post ketamine injection (15 mg/kg), but no evidence for increased cell proliferation in this relatively short experimental time frame. Although the observation of enhanced ERK1/2 phosphorylation 24 h after ketamine administration indicates a fundamental first step towards the induction of cell proliferation as activation of pERK1/2 is a key player for neuronal survival and proliferation [38,42], in future studies, neuronal proliferation will actually have to be confirmed seven days post-treatment.

Bioinformatic analysis of RNA-Seq data identified distinct differential expression of 60 genes 24 h after ketamine treatment. As confirmed by qPCR, two of the upregulated genes were p11 and IGF2, which have both been suggested to be involved in the pathology of depression and the therapeutic response of market drugs on the other hand [13,15]. As decreased p11 expression in the hippocampus of rodents with depression-like phenotype is associated with reduced cell proliferation and a subsequent upregulation after three weeks of antidepressant treatment with fluoxetine correlates with a patients recovery, accumulating evidence hints at p11 to be involved in antidepressant-induced hippocampal neurogenesis [15,16]. Furthermore, p11 levels were restored 72 h after ketamine administration, and the sustained antidepressant effect of ketamine was abolished in rats with knockdown of hippocampal p11 [43]. However, despite these astonishing findings, it is still unknown whether the neurogenic effects of ketamine depend on enhanced p11 expression. Therefore, the involvement of p11 in the proliferative effects of ketamine on NPCs has to be examined in further studies by knocking down p11 and exploring the effects on cell proliferation.

Moreover, in agreement with earlier results from Bracko et al. [12], we demonstrated that proliferation of NPCs was reduced after IGF2 knockdown. In addition, our data indicate that IGF2 treatment induces proliferation of NPCs and significantly prevented the reduction of cell proliferation upon IGF2 knockdown (Figure 6C, Appendix A), confirming the role of IGF2 as a regulator of NPC proliferation [12]. A recent in vivo study supports our findings, showing upregulation of IGF2 by ketamine (10 mg/kg) in mouse hippocampus 24 h after treatment and, furthermore, a significant decrease of the antidepressant effects of ketamine upon an IGF2 knockdown in mice [44]. Interestingly, IGF2 administration is able to increase the AMPA receptor subunit GluA1 expression in neurons, a mechanism which seems to be essential for the rapid antidepressant effects of ketamine, as pre-treatment with the AMPA receptor antagonist 2,3-dihydroxy-6-nitro-7-sulphamoyl-benzo(f)quinoxaline (NBQX) prevents the antidepressant response [21,45,46].

To further evaluate the NMDA-independent molecular effect of ketamine on NPCs, we investigated the potential involvement of cAMP signaling, as several studies observed impaired cAMP pathway in patients suffering from depression and a link between cAMP activation and neurogenesis [18]. Most importantly, a recently published study has proposed a link between cAMP signaling and the antidepressant effect of ketamine in an NMDA receptor-independent manner [37]. Consistent with the study of Wray et al. (2018), ketamine was able to increase cAMP accumulation in iPSC-derived NPCs within 15 min. As cAMP signaling is also suggested to be involved in neurogenesis and NPC cell proliferation [47], we investigated whether the cAMP pathway plays a role in the proliferative effects of ketamine. To this end, when blocking cAMP signaling using the PKA-inhibitor cAMPS-Rp, we observed a decrease of ketamine’s effect on cell proliferation, whereas PKA-inhibition alone showed no effect on proliferation, indicating that ketamine’s effect on NPC proliferation might be cAMP-dependent. This is supported by the observation that the induction of IGF2 mRNA expression by ketamine was reduced after PKA-inhibition.

In conclusion, we speculate that cAMP signaling leading to enhanced IGF2 expression may have distinct effects on cell proliferation of human NPCs. However, the molecular link between cAMP-IGF2 signaling in regard to cell proliferation needs to be further explored in subsequent studies. Our data implicate that cAMP accumulation is essential for the increase of IGF2 mRNA expression and proliferative effects of ketamine. As cAMP signaling is known to induce transcriptional changes via cAMP-PKA-CREB pathway and IGF2 and BDNF are known downstream targets of the cAMP-responsive-element binding protein (CREB), further studies will investigate the correlation of ketamine-induced cAMP signaling and downstream mechanisms regulating IGF2, p11 and BDNF signaling leading to potential neurogenesis-dependent antidepressant effects [13,47].

## 5. Conclusions

Altogether, human iPSC-derived NPCs provide a powerful model to study non-NMDA mediated actions of ketamine. Furthermore, our findings provide insight into a potentially novel mechanism responsible for ketamine’s antidepressant effect by inducing cAMP-IGF2 mediated cell proliferation of neuronal progenitor cells. Finally, further studies are essential to elucidate the molecular involvement of cAMP signaling in p11 and BDNF dependent regulation of proliferation of NPCs upon ketamine treatment, as both are suggested to play an important role in mood disorders and neurogenesis [15,37,43].

## Figures and Tables

**Figure 1 cells-08-01139-f001:**
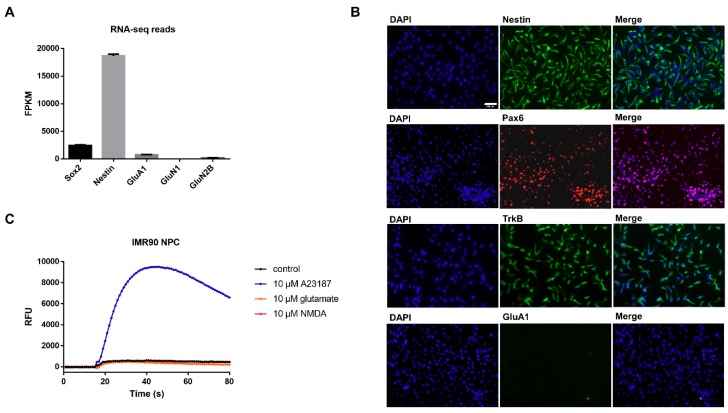
Characterization of human induced pluripotent stem cell-derived NPCs. (**A**) Read counts (fragments per kilobase of transcript per million mapped reads, FPKM) of RNA-seq analysis indicate mRNA expression of Sox2, Nestin, GluA1 (AMPA receptor subunit), GluN1 and GluN2B (NMDA receptor subunits). The data represent the means of three independent samples, and error bars were calculated using ±SEM. (**B**) Immunocytochemical characterization of iPSC-derived NPCs showing protein expression of the neuronal progenitor markers Nestin and Pax 6, and the BDNF receptor TrkB, but no expression of the ionotropic glutamate receptor AMPA-R (GluA1 subunit), scale = 100 µm. (**C**) Functional analysis of NMDA-receptors in human iPSC-derived IMR90 NPCs using the Fluo-8 calcium mobilization assay. The calcium ionophore A23187 served as a positive control. No functional NMDA receptors are expressed in undifferentiated NPCs. Abbreviations: Sox2 (sex determining region Y)-box 2), GluA1 (glutamate ionotropic receptor AMPA type subunit 1), GluN1 (glutamate ionotropic receptor NMDA type subunit 1), GluN2B (glutamate ionotropic receptor NMDA type subunit 2B), Pax6 (paired box 6), BDNF (brain derived neurotrophic factor), TrkB (tropomyosin-related kinase B).

**Figure 2 cells-08-01139-f002:**
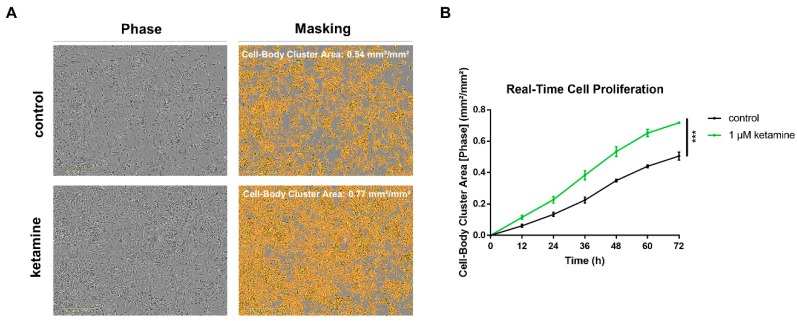
Ketamine increased cell proliferation of human iPSC-derived IMR90 NPCs. (**A**) Automated phase-contrast image segmentation using the IncuCyte^®^ NeuroTrack Software after 72 h treatment, scale = 300 µm. Confluency of cells was determined with IncuCyte^®^ NeuroTrack Software indicated as Cell-Body Cluster Area. (**B**) Phase-contrast imaging was performed using the IncuCyte^®^ Zoom time-lapse microscopy system at 37 °C for a period of 72 h. NPCs were treated with either 1 µM ketamine or 0.01% dimethyl sulfoxide (DMSO) control. The data represent means of three independent experiments, and the error bars were calculated using ± SEM.

**Figure 3 cells-08-01139-f003:**
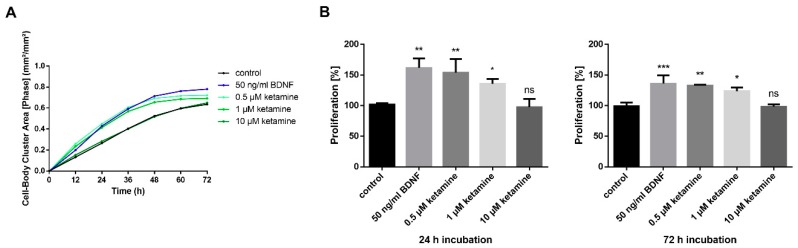
Concentration-dependent effects of ketamine on the proliferation in Ro-iPSC NPCs. (**A**) Real-time quantitative analysis of cell proliferation of an additional NPC cell line. Confluency of cells was determined with IncuCyte^®^ software indicated as Cell-Body Cluster Area. NPCs were treated with different concentrations of ketamine (0.5; 1 and 10 µM) or 50 ng/mL BDNF as a positive control. (**B**) The effect on proliferation is shown at the selected time points of 24 and 72 h compared to 0.01% dimethyl sulfoxide (DMSO) control, respectively. The data represent the means of three independent experiments. Error bars were calculated using ±SEM, and p-values were calculated against the DMSO control.

**Figure 4 cells-08-01139-f004:**
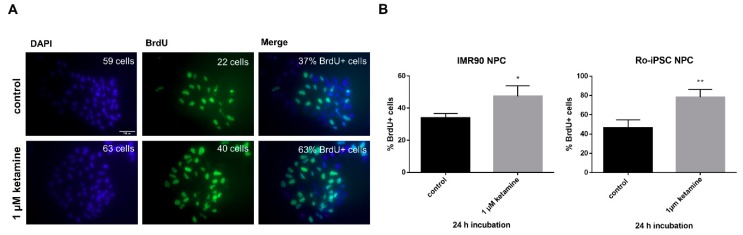
Ketamine induction of proliferation detected by 5-bromo-2′-deoxyuridine (BrdU) staining in NPCs. (**A**) Representative images of BrdU+ labeled Ro-iPSC NPCs, scale = 200 µm. NPCs were treated with 1 µM ketamine or 0.01% dimethyl sulfoxide (DMSO) control for 24 h, before cells were incubated with 10 µM BrdU labeling solution for an additional 4 h. (**B**) A histogram showing BrdU-expressing cells. BrdU incorporation was quantified by the calculation of BrdU+ cells as a percentage of the total number of 4′,6-diamidino-2-phenylindole (DAPI)-labeled nuclei using ImageJ (Fiji) for image analysis. Data represent the means of three independent experiments. Error bars were calculated using ±SEM. p-values were calculated against the 0.01% dimethyl sulfoxide (DMSO) control using the unpaired two-tailed t-test.

**Figure 5 cells-08-01139-f005:**
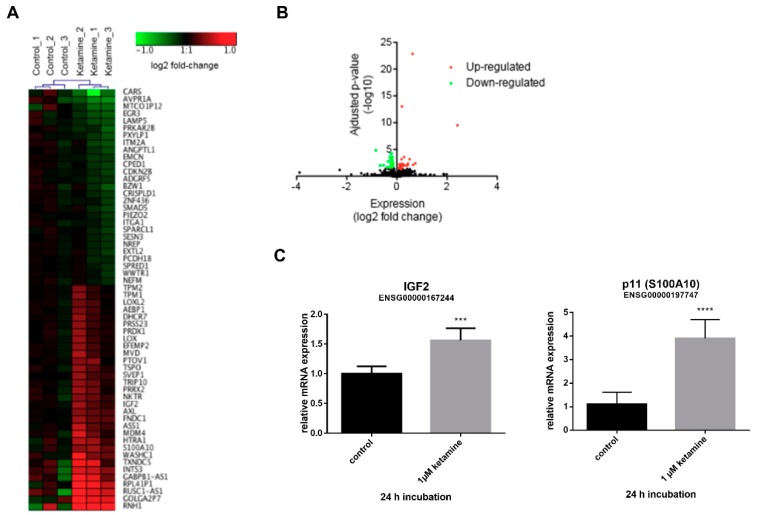
Differential gene expression in IMR90 NPCs after ketamine treatment. (**A**) RNA-Seq results are presented in a heat map showing differential mRNA expression. NPCs were treated with 1 µM ketamine or dimethyl sulfoxide (DMSO) control for 24 h (n = 3). (**B**) A volcano plot showing differentially expressed mRNAs. Thirty-one genes were upregulated (red), and 29 genes were downregulated (green) after ketamine treatment. (**C**) Confirmation of RNA-Seq data for insulin-like growth factor 2 (IGF2) and p11 by qPCR. Expression values were calculated according to the 2^−ΔΔCT^ method. Data represent the means of three independent experiments. Error bars were calculated using ±SEM. *p*-values were calculated against the 0.01% DMSO control.

**Figure 6 cells-08-01139-f006:**
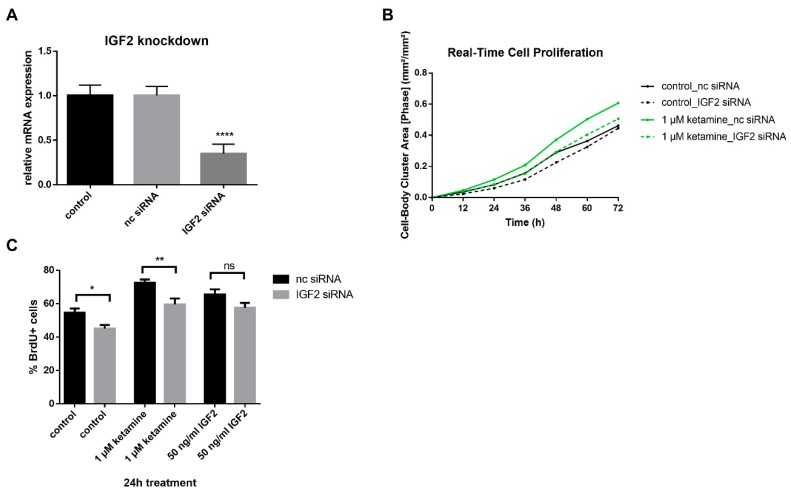
NPC cell proliferation is reduced after IGF2 knockdown. (**A**) Confirmation of IGF2 knockdown by qPCR analysis at 24 h. Expression of IGF2 was reduced by up to 66%. (**B**) The representative time-response curve of proliferating cells showing a reduction of ketamine’s effect on proliferation after IGF2 knockdown. Cells were treated with 2 µM IGF2 siRNA or nc siRNA for 24 h. After washout, ketamine was added and confluency of cells was determined with the IncuCyte^®^ NeuroTrack Software indicated as the Cell-Body Cluster Area. (**C**) An additional method showing the reduction of cell proliferation after IGF2 siRNA knockdown. After the knockdown, cells were treated with 1 µM ketamine, 50 ng/mL IGF2 or 0.01% dimethyl sulfoxide (DMSO) control for 24 h, before 10 µM BrdU labeling solution was added for an additional 4 h. The histogram shows BrdU-expressing cells. BrdU incorporation was quantified by the calculation of BrdU+ cells as a percentage of the total number of 4′,6-diamidino-2-phenylindole (DAPI)-labeled nuclei using ImageJ (Fiji) for image analysis. The data represent the means of three independent experiments. Error bars were calculated using ±SEM. p-values were calculated against the 0.01% DMSO control.

**Figure 7 cells-08-01139-f007:**
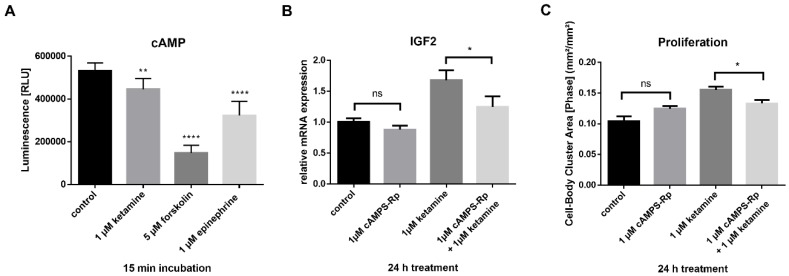
cAMP-IGF2 signaling is involved in ketamine’s proliferative effect. (**A**) Ketamine increased cAMP levels within 15 min. Intracellular cAMP levels were determined using the cAMP-Glo™ assay. Luminescence is inversely proportional to cAMP levels. IMR90 NPCs were treated for 15 min with the indicated compounds. Forskolin (adenylyl cyclase activator) and epinephrine (an agonist of adrenergic receptors) were used as positive controls. (**B**) Induction of IGF2 mRNA expression by ketamine was reduced after PKA inhibition. Cells were pretreated with 1 µM cAMPS-Rp for 15 min before ketamine or dimethyl sulfoxide (DMSO) control was added. (**C**) Inhibition of intracellular cAMP signaling reduced the effect of ketamine on proliferation. Effect on proliferation is shown at the selected time point of 24 h compared to DMSO control (n = 3). The data represent the means of three independent experiments. Error bars were calculated using ±SEM. p-values were calculated against DMSO control.

**Figure 8 cells-08-01139-f008:**
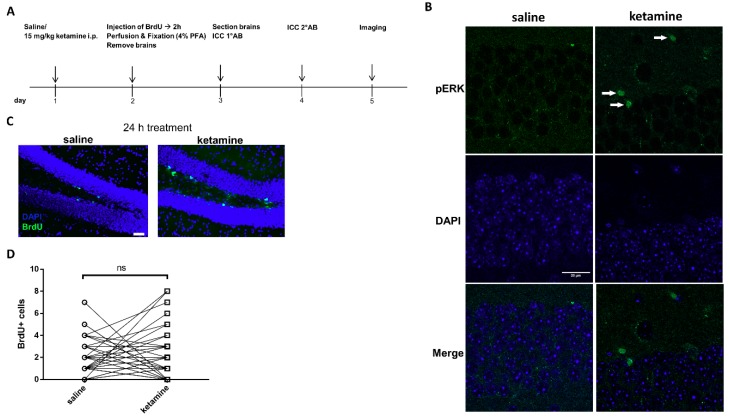
Ketamine increased extracellular signal-regulated protein kinases 1 and 2 (ERK1/2) phosphorylation in mice 24 h after administration (**A**) Experimental timeline: male C67BL/6 mice were injected intraperitoneal (i.p.) with 15 mg/kg ketamine or 0.9% saline. 24 h after injection, mice were injected with 200 mg/kg BrdU and, 2 h later, anesthetized and perfused with 4% paraformaldehyde PFA) and prepared for immunohistochemistry. (**B**) Representative images showing that mice display enhanced ERK1/2 phosphorylation 24 h after acute ketamine treatment, scale = 20 µm. (**C**) Representative images of BrdU staining 24 h after treatment, scale = 50 µm. (**D**) BrdU+ cells in hippocampal sections (40 µm) quantified as the absolute number of BrdU+ cells in each section (four mice for each condition and 12 sections per mouse; One-way ANOVA, posthoc t-test).

**Figure 9 cells-08-01139-f009:**
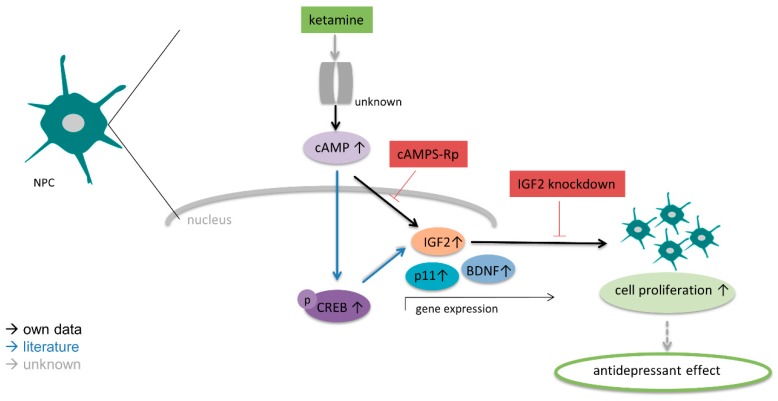
Hypothesis of the molecular mechanism of action in human NPCs induced by ketamine. Ketamine treatment increases IGF2 expression via cyclic adenosine monophosphate (cAMP) signaling leading to enhanced NPC proliferation.

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
