# Peer review of "Ketamine Increases Proliferation of Human iPSC-Derived Neuronal Progenitor Cells via Insulin-Like Growth Factor 2 and Independent of the NMDA Receptor"

_cells, 2019, doi:10.3390/cells8101139_

Round 1
Reviewer 1 Report
1) Please, perform additional experiment the detection of NR1 (not NR2B), for critical functional NMDAR subunit.
2) Please, examine Mg-free condition in Tyrode's buffer for Calcium rise assay. NMDAR are blocked Mg++.
3) Please add to the result or the legend about what kind of statistics you did.
4) Please add control nc-siRNA to fig.6 A and C. and do two-way ANOVA.
5) If NMDAR independent mechanism, Please add serotonin receptor etc. as a candidate other than NMDAR as a ref.
Reviewer 2 Report
In this study the authors assess the mode of action of ketamine-induced increase of proliferation of human iPSC-derived neuronal progenitor cells. They conclude that the phenomenon is IGF-2-regulated and NMDA receptor-independent.
The manuscript contains some grammar and syntax errors that need to be corrected (for examples, please refer to the uploaded file).
In addition, there are some issues that need to be resolved before the study can be considered for publication.
Some experiments of the study were conducted in IMR90 and some in Ro-iPSC NPCs without an explanation for this selection to the reader. I suggest that all experiments presented are performed in both cell lines for reasons of uniformity and coherence. Otherwise, the authors should present a panel of Figures from experiments performed in one of the two cell lines (always the same one) and confirm their findings in the other cell line in some cases (and probably refer to this data as data not shown). Sentence in ln100-101 could be deleted because the reader cannot understand here why the authors selected this range of ketamine concentrations (which is explained later in the text). In the Materials and Methods section some parts in paragraph 2.5 need clarification (please refer also to the uploaded file). Do the authors believe that incubation time with BrdU for 4 or 2 hours is sufficient to allow incorporation in the DNA during novel DNA synthesis? If the authors extended this incubation time, they may result in greater differences. The authors should clarify if they used the same RNA samples for RNAseq analysis and verification experiments. Why the concentration of 15 mg/kg was selected for in vivo experiments since in previous published reports other concentrations had been used? The order of Figures is sometimes different from their appearance in the text (please also refer to the uploaded file). This should be corrected. The authors should elaborate on the translation of RNAseq analysis data in Fig. 1 (please also refer to the uploaded file). In Figure 3, can you explain why the percentage at 1 μΜ is smaller than that at 0.5 μΜ? And since 0.5 μM was more effective, why did the authors select 1 μM for the rest of their experiments? In siRNA experiments the concentration of 2 μΜ is rather low. The authors could repeat the experiment with a higher concentration which could be more effective for knocking-down. Figure 8A, 9 and S5 are not mentioned in the text. This should be corrected.
Reviewer 3 Report
In this study, Grossert et al. investigated the effects of ketamine on the proliferation of human iPSC-derived NPCs. They demonstrated that ketamine induce the proliferation of human iPSC-derived NPCs by the induction of IGF2 expression via NMDA receptor-independent manner. Although this manuscript contains interesting results, there are some issues.
1. Data for time course of IGF2 expression is necessary.
2. The authors must investigate protein levels of IGF2.
3. IPSC-derived NPCs proliferate by the addition of IGF2? In Fig. 6B, the authors should investigate whether addition of IGF2 cancels the effects of siRNA knockdown.
4. In Fig. 7, the effects of ketamine on cAMP production were very weak. It is plausible to the involvement of other pathway.
Round 2
Reviewer 1 Report
If possible, use multiple tests instead of t-test with statistical processing.
Author Response
We would like to thank the reviewer for his/her final feedback which we have addressed (in blue color).
We look forward to a favorable response.
Sincerely,
Nicole Teusch
Reviewer 1:
If possible, use multiple tests instead of t-test with statistical processing.All statistical analysis and graphs were generated with GraphPad Prism v. 7.02 (Graphpad Software, San Diego, CA, USA) using one-way ANOVA and the underlying Sidak’s multiple comparisons test or unpaired two-tailed t-test for the comparison of two groups. As suggested, in the legends of Figure S5 the information regarding the statistical analysis has been corrected (as highlighted in green color).

Reviewer 2 Report
This is the revised version of a previously submitted work.
The authors have addressed most of my concerns raised during the previous round of the reviewing process.
Author Response
We would like to thank the reviewer for the constructive and favourable feedback. Minor spell check has been performed to the best of our knowledge.
Reviewer 3 Report
Authors adequately revised the manuscript following my comments.
Author Response
We would like to thank the reviewer for the constructive and favourable feedback.